# Research on Damage and Deterioration of Fiber Concrete under Acid Rain Environment Based on GM(1,1)-Markov

**DOI:** 10.3390/ma14216326

**Published:** 2021-10-23

**Authors:** Jianqiao Yu, Hongxia Qiao, Feifei Zhu, Xinke Wang

**Affiliations:** 1School of Civil Engineering, Lanzhou University of Technology, Lanzhou 730050, China; 20150207024@m.scnu.edu.cn (H.Q.); ff83zhu2017@163.com (F.Z.); wangxinke9611@163.com (X.W.); 2Western Ministry of Civil Engineering Disaster Prevention and Mitigation Engineering Research Center, Lanzhou University of Technology, Lanzhou 730050, China

**Keywords:** fiber concrete, acid rain corrosion, damage deterioration, GM(1,1)-Markov

## Abstract

With steel fiber and basalt fiber volume dosing serving as variation parameters, a total of 200 d cycles of acid rain corrosion cycle tests were conducted on fiber concrete in this study. We selected three durability evaluation parameters to assess the degree of damage deterioration on fiber concrete, used scanning electron microscopy, mercury intrusion porosimetry, and a dimensional microhardness meter to analyze the concrete micromorphology, and established a GM(1,1)-Markov model for life prediction of its durability. Results reveal that the acid rain environment is the most sensitive to the influence of the relative dynamic elastic modulus evaluation parameter, and concrete has specimens that show failure damage under this parameter evaluation. Incorporation of fibers can reduce the amount of corrosion products inside the concrete, decrease the proportion of harmful pores, optimize the mean pore-size, and significantly improve the resistance to acid rain attack. Concrete with 2% steel fiber and 0.1% basalt fiber by volume has the least change in durability damage, and the predicted service life by GM(1,1)-Markov model is 322 d.

## 1. Introduction

In recent years, with the dramatic increase in emissions of nitrogen oxides and sulfides, acid rain pollution has become one of the important environmental issues in the world today, and it shows a continuous expanding trend of the affected area. Research reports indicate that over 100 billion yuan of annual losses in China is caused by acid rain problems, with the largest proportion of economic losses done by acid rain corrosion of construction materials such as concrete [1]. The acid rain in China, largely of sulfuric acid type, is mainly distributed in the south of the Yangtze River, where the service life of concrete structure buildings is significantly lower than that of non-acid rain areas, which requires frequent repair to extend the service life. Sulfuric acid type acid rain can react with alkaline compounds in concrete, changing the cementite composition, which will make the concrete material expand in volume and crack, and eventually lead to the failure of the concrete structure. Therefore, there is an urgent need for an effective method to inhibit the corrosion of concrete materials by acid rain [2,3,4].

At present, researchers have conducted some research on the mechanism of acid rain corrosion of concrete and the deterioration of concrete performance after corrosion, and proposed improvement measures to reduce the corrosion of concrete by acid rain. The improvement measures mainly lie in two aspects: on the one hand, the incorporation of mineral admixtures can improve the corrosion resistance of concrete to acid rain. Shi et al. [5] studied the mechanism of acid rain corrosion resistance of concrete with added mineral admixtures through X-ray diffraction, scanning electron microscopy, and energy spectrometry tests, and the results showed that polypropylene fiber fly ash expanded concrete had the best acid rain corrosion resistance. Ahmad et al. [6] investigated the effect of rice husk ash on cement in an acid rain environment and found that the incorporation of 5% rice husk ash can improve the mechanical properties of concrete in this environment, and also the microstructure of cement. Wang et al. [7] pointed out that acid rain corrosion damage of concrete containing mineral admixtures is mainly the result of the combined effect of dissolution damage caused by H^+^ and expansion damage caused by SO_4_^2−^. Xu et al. [8] simulated the ultimate bearing capacity of biased columns with and without lithium slag doping under acid rain environment based on ANSYS 15.0 finite element software (Pittsburgh, PA, USA), and it was found that the doping of lithium slag had some improvement on the bearing capacity and lateral stiffness of biased columns under acid rain corrosion, and the ultimate bearing capacity of biased columns decreased with the extension of acid rain corrosion time. On the other hand, considering the influence of construction materials on engineering economy, fiber concrete has been gradually popularized in engineering, and its acid corrosion resistance has gradually attracted people’s attention. Adding steel fibers (SF) into concrete is the most widely studied and the earliest developed reinforcement method. By observing the XRD patterns of the corrosion products of steel fiber shotcrete after a sulfate attack, Niu et al. [9] found that the corrosion products were poorly crystallized and did not yet form destructive crystals, so it was demonstrated that steel fibers can reduce the number of internal microcracks in the specimens, which results in a lower concentration of internal sulfate ions. Wang et al. [10] used mercury porosimetry test to observe that steel fibers can improve the acid rain corrosion resistance of concrete by improving the pore structure and enhancing the bonding effect of the concrete matrix. Basalt fiber (BF) has good economic advantages because of its hydrophilic and flexible structure that is easy to disperse in concrete and can be produced through conventional processes and equipment. Zollo [11] believes that incorporating two or more fibers into a cementitious material can make the composite significantly better than the sum of the properties of all single fiber-reinforced cementitious materials. Zhang [12], Koksal [13], Ibrahim [14], etc., in order to further promote the application of steel fiber and basalt fiber in the construction industry, conducted a large number of tests on the compressive strength and bending toughness of steel-basalt fiber concrete, and each proposed the corresponding optimal dosing. Through the above research results, it is easy to find that the current academic research on steel-basalt fiber concrete research results is quite abundant, mainly focusing on the optimization of the mix ratio and mechanical properties. However, there are few studies on the durability performance of hybrid-fiber-reinforced concrete under an acid rain environment, which sets a new research direction and objective for the present study.

In order to ensure the safety of concrete materials in an acid rain corrosive environment, it is especially important to judge whether concrete can complete the intended function within the design reference period. The life prediction of concrete durability has become a hot topic of research in the civil engineering industry [15,16,17,18,19]. Gray system theory has the outstanding features of a simple modeling process and concise model expression, among which the GM(1,1) prediction model is the most widely used. Since the final reduced form of the conventional GM(1,1) model is an exponential function with strict monotonicity, and the damage deterioration process of concrete has great randomness and uncertainty, direct use of the GM(1,1) model may lead to certain errors in the prediction results. Markov chain theory can describe the stochastic dynamic change process of the predictor well, and predict the next state of the predictor by studying the transfer probability among the states, which can be combined with GM(1,1) to improve the prediction accuracy of the model. In view of this, in order to be closer to reality and easy to apply, the artificial climate environment accelerated corrosion technology was adopted in this paper to simulate the acid rain environment. Five groups of fiber concrete specimens were selected as the research object to study its damage deterioration law with the increase of the number of acid rain spray-light corrosion cycles. The microscopic morphology, pore structure, and mechanical properties of the interfacial transition zone (ITZ) of the corroded fiber concrete were characterized and analyzed by SEM, mercury intrusion porosimetry (MIP), microhardness (MH), and other microscopic tests, so as to establish a dynamic analysis model GM(1,1)-Markov for the life prediction of durability indexes considering the influence of acid rain damage, with a view to provide theoretical support for the assessment of durability performance of fiber concrete under an acid rain environment. This is to provide theoretical support for the durability assessment of fiber concrete in an acid rain environment.

## 2. Materials and Methods

### 2.1. Materials and Mixture Proportions

P.O 42.5 grade ordinary silicate cement (Qilianshan Cement Plant Ltd., Lanzhou, China) with good stability was used in this study. Class II fly ash (Lanzhou Thermal Power Co., Lanzhou, China) with a specific surface area of 440 m^2^/kg was also utilized, and the chemical components of the cementitious material are shown in Table 1. High-efficiency water reducing agent was obtained from Lanzhou Huajian Concrete Co. (Lanzhou, China), and the water reduction rate was about 23%. The coarse aggregate was made of crushed stone from the Lanzhou area with an apparent density of 2660 kg/m^3^ and mud content of 0.5%. The fine aggregate was made of Lanzhou Anning River sand, which was medium sand with a fineness modulus of 3.18 and an apparent density of 2581 kg/m^3^. The water used was tap water. The steel fiber (SF) and basalt fiber (BF) were provided by Shanghai Chemical Building Materials Factory (Shanghai, China), and the main physical properties are detailed in Table 2. Meanwhile, the selected fibers conform to the corresponding industry standards and current national standards. According to JGJ/T 221-2010 [20], the test design concrete strength grade was uniformly C40, and the water-cement ratio was 0.4. The details of the ratio and concrete compressive strength are shown in Table 3.

### 2.2. Experimental Methods

According to the ratios given in Table 3, concrete specimens measuring 100 × 100 × 100 mm^3^ were prepared. Each group of specimens was placed in a curing chamber at a temperature of 30 ± 2 °C and with a relative humidity of 80% for 28 days. Then, the test specimens were transferred into the indoor cyclic test chamber ZC-2512, as shown in Figure 1. To simulate the spray-light cycles in an acid rain corrosive environment, the top of the test chamber was installed with flexible nozzles which could rotate in all directions. In addition, the Philips heating lamps were also installed on the top to simulate the lighting process and adjust the test chamber temperature. The experimental parameters, such as spraying time, light time, running time per circulation, and total number of cycles, could be set through a programmable controller. In accordance with the Chinese standard for test methods of long-term performance and durability of ordinary concrete GB/T 50476-2019 [21], the spray-light accelerated corrosion regime under the acid rain environment is shown in Figure 2. To simulate the spray-light circulation and accelerate the corrosion process, a single corrosion circulation was set to consist of 16 h of spraying the acid rain solution at 35 °C, and 8 h of lighting at 60 °C. One test cycle lasted 24 h, and the total test cycle lasted 200 days. Moreover, the pH of the prepared acid rain solution was 3.0, and the specific composition is detailed in Table 4.

### 2.3. Testing Procedure

In order to effectively track the durability damage deterioration pattern of fiber reinforced concrete, the data of concrete specimens were collected at 20-day intervals. Similarly, the GB/T 50476-2019 outlines three conditions for concrete failure after spray-light cycles. When the compressive strength is reduced by 25%, the mass is reduced by 5%, or the relative dynamic elastic modulus is reduced by 40%, a condition is fulfilled, indicating that the test specimen has reached durability failure. In this test, the number of spray-light cycles that the fiber concrete withstood was used to express its resistance to acid rain environmental corrosion. The YH-300 electro-hydraulic servo press (Shanghai, China) was used for the test of compressive strength. The NM-4A ultrasonic tester (Beijing, China) was used to test the sound velocity. A high-precision electronic balance was utilized to test the mass. In addition, in order to facilitate comparison of the damage deterioration rates of fiber concrete under acid rain corrosion in different test conditions, the test data were normalized and evaluated on the basis of the compressive strength evaluation parameter ξ_1_, relative mass evaluation parameter ξ_2_, and relative dynamic elastic modulus evaluation parameter ξ_3_ for concrete durability under different acid rain corrosion ages, respectively. Each evaluation parameter is shown in Equations (1)–(3):(1)$$\xi_{1}=\frac{f_{\mathrm{cn}}{/f}_{c0}-0.75}{0.25}\times 100\%$$
(2)$$\xi_{2}=\frac{m_{n}{/m}_{0}-0.05}{0.95}\times 100\%$$
(3)$$\xi_{3}=\frac{v_{n}^{2}{/v}_{0}^{2}-0.60}{0.40}\times 100\%$$
where f_cn_ and f_c0_ are the compressive strengths (MPa) after n test cycles and at the initial stage, respectively. m_n_, m_o_ denote the mass of the specimen (kg) measured after n test cycles and at the initial stage, respectively. v_n_, v_0_ denote the ultrasonic velocity (m/s) measured after n test cycles and at the initial stage, respectively. When ξ is less than 0, the concrete is considered to have failure damage, and when ξ is between 0 and 1, the specimen is considered to have damage deterioration.

Some concrete specimens were transported back to the laboratory at the end of the 200-day test cycle to reveal the damage deterioration mechanism of fiber concrete in the acid rain environment. Samples were made, and the damage of fiber concrete was observed using SEM, mercury intrusion porosimetry, and microhardness. For the preparation of fiber concrete SEM specimens, the fiber concrete was crushed and the cement mortar pieces were taken as samples, and the sample size was approximately 1 × 1 ×1 cm^3^, which was metal sputtered before the test. SEM testing was performed using a Zeiss Sigma 300 field emission electron microscope. Mercury porosimetry specimens were made by cutting the test sample with volume size not larger than 0.5 × 0.5 × 1 cm^3^, and placed in anhydrous ethanol to terminate hydration for 48 h. The AutoPoreIV9510 mercury piezometer (Shanghai, China) was used for the mercury intrusion porosimetry test. Each microhardness test specimen was cut in the middle using a cutting machine to create a section with a thickness of 20 mm and then polished. The HVS-1000A Vickers microhardness tester (Guangdong, China) was used to test the microhardness values at different locations from the edge of the aggregate by hitting the points sequentially from the side of the aggregate toward the cement matrix. Seven hitting locations were established, with the first point being 10 µm from the edge of the aggregate, the next two points being 20 µm apart, and the last point being 130 µm from the edge of the aggregate. The microhardness value at each point location was taken as the average of five hits at the same distance from the edge of the aggregate at the arc location of the point.

## 3. Results and Discussion

### 3.1. Evaluation Parameters of Compressive Strength

With the increase in the number of spray-light corrosion cycles, the change curve of compressive strength evaluation parameters for each group is shown in Figure 3. Compressive strength increased in the early stages with an increase in age of corrosion for all groups of concrete specimens. The early growth of ξ_1_ was greater for the fiber concrete specimens than for OPC. The S2B0.1 group, in particular, reached the peak of ξ_1_ at 40 days with 122.07%, which was found to be the largest increase. After 200 corrosion cycles, the compressive strength of each group of specimens was damaged to varying degrees. In the OPC group, ξ_1_ decreased to 15.40% with the most severe impairment. By contrast, the damage of the fiber-doped specimen ξ_1_ was relatively light. The degree of damage in descending order was S2B0.2 > S1B0.2 > S1B0.1 > S2B0.1. Two main factors explained the increase in the compressive strength of concrete in the early stages of corrosion by the acid rain environment. First, the specimen continued to hydrate in the corrosion solution. Second, the corrosion products that were generated in the early stages filled the internal pores of the concrete, thereby making the concrete increasingly dense and causing the compressive strength to rise. However, as the age of corrosion increased, the large number of generated expansions produced microcracks by extrusion within the concrete, thus causing a decrease in compressive strength. However, as the corrosion age increased, the SO_4_^2−^ in acid rain solution gradually migrated into the concrete, and when the internal pores did not have enough space to accommodate the expansion products, micro-cracks formed quickly, resulting in a decrease of the compressive strength [22]. Meanwhile, fibers made the concrete increasingly dense inside and slowed down the damage to the concrete matrix caused by corrosion ions [23]. Thus, the ξ_1_ corrosion age variation curve of the concrete specimens mixed with fibers was more stable than that of OPC.

### 3.2. Evaluation Parameters of Relative Mass

Figure 4 shows the relationship between the relative quality evaluation parameter ξ_2_ of fiber concrete with acid rain corrosion time. The data sets showed a synchronous upward change at 20 days, followed by a fluctuating change with the increasing age of corrosion ξ_2_ and a synchronous decrease at 160 days. The highest mass loss rate was observed in the OPC group at 200 days when ξ_2_ was 70.4%. The mass loss of the concrete specimens mixed with fibers was not significant, with only 5% of ξ_2_ of S2B0.1 being lost. This result indicated that the incorporation of SF and BF could significantly improve the mass loss inhibition ability of concrete. In the middle of the test, ξ_2_ for each group of data fluctuated because of the dynamic changes in the filling effect of corrosion products and the destructive effect of corrosion expansion crystals. With the passage of time, the filling effect of corrosion products gradually transformed into expansion damage. The specimen surface began to show considerable peeling and slagging, and the macroscopic performance indicated that ξ_2_ began to decline. In addition, ξ_2_ of fiber concrete at 200 days was greater than ξ_1_, which may be due to the fact that the physical crystallization destruction ability of spray-light has not yet caused the fiber concrete to exhibit quality loss behaviors, including dissolution precipitation of cementitious materials and aggregate shedding.

### 3.3. Evaluation Parameters of Relative Dynamic Elastic Modulus

The experimental data for the relative dynamic elastic modulus evaluation parameter ξ_3_ of fiber concrete are shown in Figure 5. The ξ_3_ of OPC rose slowly at the early stage of corrosion and decreased rapidly after 40 days. ξ_3_ decreased to 0 at 180 days when the concrete material’s durability failed. Relative to that of plain concrete, ξ_3_ improved to different degrees for each group of specimens of fiber concrete. At 200 days, the ξ_3_ sizes of the fiber concrete specimens were S2B0.1 > S2B0.2 > S1B0.1 > S1B0.2. ξ_3_ of S1B0.2 dropped to 0, which reached the damage criterion. The reason for this may be due to the increase in the number of cycles, as the spray-light-generated power drove the migration of corrosion ions into the concrete, and constantly consumed the gelatinous C-S-H and C-A-H, resulting in the internal pore wall of the concrete to easily crack under expansion stress. Moreover, the propagation speed of ultrasound weakened, and the ξ_3_ value decreased. SF and BF worked together with C-S-H to form a fibrous mesh bonding structure to enhance the adhesive properties between the cementitious materials and the aggregates and share the expansion stresses generated within the concrete caused by acid rain corrosion. These conditions led to the inhibition of the decrease in dynamic elastic modulus. By contrast, the increase of the SF admixture caused the ξ_3_ of concrete to rise under the same BF admixture. The SF obviously exerted a great influence on the improvement of the relative dynamic elastic modulus of concrete. Poorsaheli et al. [24] concluded that a high modulus of elasticity and high strength SF can act as a skeleton in concrete to play a bridging role and that a wavy appearance can enhance the mechanical bite with concrete and fully improve the denseness of the internal structure of concrete; such improvement exerts a positive impact on the improvement of the dynamic modulus of elasticity. On the basis of the combination of ξ_1_, ξ_2_, and ξ_3_, S2B0.1 was found to be the best mixture proportion for this test. The order of the magnitude of sensitivity of the three durability evaluation parameters of fiber concrete under acid rain corrosion was ξ_3_ > ξ_1_ > ξ_2_. This result is explained as follows: at the end of the test, durability failure was noted in the specimens with ξ_3_ of 0 in each group; those with ξ_1_ and ξ_2_ still exceeded 0.

### 3.4. Microstructure Investigation

Figure 6 shows the SEM images of specimens in each group after 200 days of acid rain corrosion. As depicted in Figure 6a, OPC generated a large number of lumpy and short rod-shaped corrosion products. The corrosion products were porous, uneven, and densely distributed, thereby causing a large number of cracks in the internal structure of concrete and providing a channel for the invasion of corrosion media. The pores in the cracks further provided an environment for the growth of swelling crystals, such as large calcium alumina. The crystals eventually generated concentrated stresses within the pore walls, thus leading to the extensive cracking of the concrete and resulting in durability damage [25]. Figure 6b shows traces of fiber pull-out at the interface of S1B0.1 and obvious multicracks inside the concrete, but the crack width appeared to be relatively reduced. Massive crystals grew out of the interior at the fractures, and bar crystals were observed to be developing. The incorporation of SF and BF played a role in weakening the internal expansion force caused by the corrosion products, changing the stress distribution state, and inhibiting the cracking of concrete to a certain extent. As shown in Figure 6c, the interface of S1B0.2 was loosely bonded to the concrete, with a large number of filamentary and granular corrosion products scattered on the fiber surface and not forming a tight whole. This result indicated that the BF doping was too large and thus affected the adequate bonding of the fibers to the matrix and increased the internal defects of the specimen. As depicted in Figure 6d, the interfacial structure of S2B0.1 was relatively dense, and dense C-S-H gels were observed. The fibers were uniformly oriented and distributed in the matrix, which acted as a secondary micro-strengthening. At the same time, the fiber interaction formed a three-dimensional mesh structure, and the bridging effect of the fiber strengthened the bond between the fiber and the matrix, thereby effectively enhancing the concrete’s ability to resist acid rain corrosion [26]. As shown in Figure 6e, S2B0.2 had a large crack generation because of the corrosion of SF by the corrosive ions that made the fiber volume increase and led to the concrete cracking.

### 3.5. Pore Structure

According to Dong’s study [27], concrete pore size can be divided into four categories: harmless hole (<20 nm), less harmful hole (20–50 nm), harmful hole (50–200 nm), and more harmful hole (>200 nm). Moreover, concrete damage is mainly caused by the increase in the proportion of harmful holes and multi-harmful holes in the pore transformation process. 

Figure 7 shows the pore size distribution of fiber concrete after 200 days of exposure to the acid rain environment. The internal harmful pores and multi-harmful pores of OPC accounted for a significantly higher proportion than those of the other groups of specimens, thus indicating that the doping of fibers inhibited the swelling effect of the corrosion products on the pores and converted the large pores into less harmful pores with small sizes. The number of harmless pores of the fiber concrete was ranked as follows: S2B0.1 > S2B0.2 > S1B0.1 > S1B0.2. The comparison showed that the increase of BF doping decreased the proportion of harmless pores when the SF doping was fixed. This result is attributable to the fact that BF is water-repellent material; a relatively high dose of BF in the cement slurry easily agglomerates a small number of harmful pores [28]. The cumulative growth of large quantities of salt corrosion products in these pores produces uniform stresses that increase the rate of conversion of harmless pores into harmful and multi-harmful pores.

Figure 8 shows the pore size differential distribution curves of specimens in each group after 200 days of acid rain corrosion. The peak of the differential curve reflected mean pore-size, which represented the most developed pore size range in the tested material [29]. Meanwhile, for the groups of specimens doped with fibers, the mean pore-size was smaller than those of OPC. The mean pore-size of S2B0.1 was 21 nm, which was much smaller than that of OPC (66 nm). This result indicated that the pore size generated in the fiber concrete was concentrated in the range of less harmful pores and that the pore structure was increasingly dense. Such a condition is conducive to the resistance of the transport of acid rain solution and the invasion of corrosive ions in the system. However, S1B0.2 showed a sudden increase in mercury feed near 208 nm, which was in the range of multi-harmful holes. Therefore, an excessively high dose of BF exerted a negative effect on the pore structure.

### 3.6. Microhardness Analysis

Figure 9 shows the microhardness of specimens in each group after 200 days of acid rain corrosion. When the test indenter gradually transitioned from the edge of the aggregate to the cement paste interface, the microhardness values showed an overall gradient change. The aggregate−cement paste interface transition zone (ITZ) of OPC can be classified to be 68 µm according to the variation of microhardness values. Meanwhile, the ITZ of fiber concrete was significantly thin, with S2B0.1 showing a thickness of 27 µm. The thickness of the ITZ is related to the degree of enrichment of corrosion products; the more significant the enrichment is, the greater the thickness of the ITZ will be. After the 200-day cycle test, the interfacial mechanical properties of the specimens in each group were ranked as: S2B0.1 > S1B0.1 > S1B0.2 > S2B0.2 > OPC, which was consistent with the corresponding macroscopic mechanical properties. The results revealed that fibers could provide sufficient hydration reaction time inside concrete by inhibiting the intrusion of corrosion ions, substantially eliminating the weak link of the ITZ, and improving the microstructure of the interface zone [30]. In addition, the right amount of BF can be fully filled in the tiny pores of the ITZ to play a cementing role. The combined effect of the above two phenomena leads to the reduction of microcracking in the ITZ area of fiber concrete, which is expressed as an increase in ITZ microhardness.

## 4. GM(1,1)-Markov Prediction Model

### 4.1. Derivation of the GM(1,1)-Markov Model

A GM(1,1) prediction model is established, and let the original data series be as follows:(4)$$X^{\left(0\right)}\left(t\right)=\left\{X^{\left(0\right)}{(1),X}^{\left(0\right)}(2),\ldots X^{\left(0\right)}\left(n\right)\right\}$$

One of the important data operations in GM(1,1) model is the accumulative generating operation, abbreviated as AGO. Then, X^(1)^(t) is considered as the 1-AGO generation series of X^(0)^(t) as follows:(5)$$X^{\left(1\right)}\left(t\right)=\left\{\sum_{i=1}^{1}X^{\left(0\right)}(i),\sum_{i=1}^{2}X^{\left(0\right)}(i),\ldots ,\sum_{i=1}^{n}X^{\left(0\right)}\left(i\right)\right\}$$

Define $Z^{\left(1\right)}=\left\{Z^{\left(1\right)}(2),Z^{\left(1\right)}(3),\ldots ,Z^{\left(1\right)}(n)\right\}$ as the background value with respect to X^(1)^(t), where Z^(1)^(t) is calculated as follows:(6)$$Z^{\left(1\right)}\left(t\right)=0.5\times \left(X^{\left(1\right)}\left(t\right)+X^{\left(1\right)}(t+1)\right)$$

This formula leads to the whitening differential equation for the GM(1,1) model, as shown in Equation (7).
(7)$$\frac{{\mathrm{dX}}^{\left(1\right)}}{\mathrm{dt}}+{\mathrm{aX}}^{\left(1\right)}=u$$
where a is the development coefficient, and u is the gray input coefficient. The values of a and u depend on the form of the original sequence and the construction of the background value. The values of the coefficients a and u can be determined using Equations (8)–(10).
(8)$$\left[\begin{array}{l} a\\ u \end{array}\right]={\left(B^{T}B\right)}^{-1}B^{T}Y_{n}$$
(9)$$B=\left[\begin{array}{cc} -Z^{\left(1\right)}\left(1\right) & 1\\ \begin{array}{c} -Z^{\left(1\right)}\left(2\right)\\ \vdots \\ -Z^{\left(1\right)}\left(n\right) \end{array} & \begin{array}{l} 1\\ \vdots \\ 1 \end{array} \end{array}\right]$$
(10)$$Y_{n}=\left[\begin{array}{c} X^{\left(0\right)}\left(2\right)\\ X^{\left(0\right)}\left(3\right)\\ \vdots \\ X^{\left(0\right)}\left(n\right) \end{array}\right]$$

The GM(1,1) prediction model and GM(1,1) prediction value can be obtained by substituting the solved coefficients a and u into Equation (7) and solving the whitening differential equation as follows:(11)$$\left\{ \begin{array}{l} {\hat{X}}^{\left(1\right)}\left(t\right)=\left(X^{\left(0\right)}\left(1\right)-\frac{u}{a}\right)e^{-a(t-1)}+\frac{u}{a}\\ {\hat{X}}^{\left(0\right)}\left(t\right)={\hat{X}}^{\left(1\right)}{(t)-\hat{X}}^{\left(1\right)}\left(t-1\right) \end{array}\right.$$

The residual prediction model is then established. As a result of the existence of the negative values of the original values of the residual ε, the residual ε^(o)^(t) is obtained after absolute value transformation.
(12)$$\varepsilon^{\left(0\right)}\left(t\right)=\left|\varepsilon \right|=\left|X^{\left(0\right)}{(t)-\hat{X}}^{\left(0\right)}\left(t\right)\right|$$

Then, the residual series can be created as follows:(13)$$\varepsilon^{\left(0\right)}\left(t\right)=\left\{\varepsilon^{\left(0\right)}{(2),\varepsilon }^{\left(0\right)}(3),\ldots \varepsilon^{\left(0\right)}\left(n\right)\right\}$$

By repeating steps (5) to (10), the residual prediction model and the residual prediction values are obtained as follows:(14)$$\left\{ \begin{array}{l} {\hat{\varepsilon }}^{\left(1\right)}\left(t\right)=\left(\varepsilon^{\left(0\right)}\left(2\right)-\frac{u}{a}\right)e^{-a(t-1)}+\frac{u}{a}\\ {\hat{\varepsilon }}^{\left(0\right)}\left(t\right)={\hat{\varepsilon }}^{\left(1\right)}{(t)-\hat{\varepsilon }}^{\left(1\right)}\left(t-1\right) \end{array}\right.$$

By analyzing the regularity information in the residuals, the Markov transfer matrix can be established to specify their states: state 1 when the residuals are positive and state 2 when they are negative. The state transfer probabilities can be obtained according to the positivity and negativity of the states as follows: (15)$$P_{\mathrm{ij}}=\frac{M_{\mathrm{ij}}}{M_{i}},\;i=1,2;j=1,2$$
where: P_ij_ is the transfer probability from state i to state j; M_ij_ is the transition time from state i to state j, and M_i_ is the amount of data belonging to the i-th state.

The state probability transfer matrix can be derived from the state transfer probabilities as follows:(16)$$P=\left[\begin{array}{cc} P_{11} & P_{12}\\ P_{21} & P_{22} \end{array}\right]$$

The final GM(1,1)-Markov model and the predicted values are obtained as follows:(17)$$\left\{ \begin{array}{l} {\hat{Y}}^{\left(1\right)}\left(t\right)={\hat{X}}^{\left(1\right)}\left(t\right)+1_{\left\{P(+)\geq P(-)\right\}}{\hat{\varepsilon }}^{\left(1\right)}\left(t\right)-1_{\left\{P(+)<P(-)\right\}}{\hat{\varepsilon }}^{\left(1\right)}\left(t\right)\\ {\hat{Y}}^{\left(0\right)}\left(t\right)={\hat{Y}}^{\left(1\right)}{(t)-\hat{Y}}^{\left(1\right)}\left(t-1\right) \end{array}\right.$$

### 4.2. Analysis of Prediction Result

From the durability test results in Section 3, it can be seen that each group of specimens is most sensitive to the acid rain environment with the relative dynamic elastic modulus E_r_ as the durability evaluation parameter. The analysis of Equation (3) shows that when it is lower than 0.6, the concrete specimen is considered to have failure damage. In this section, the relative dynamic elastic modulus damage degradation process of OPC is selected as the input data and the GM(1,1)-Markov model of OPC under acid rain environment is developed as follows:(18)$$\left\{ \begin{array}{l} {\hat{Y}}^{\left(1\right)}\left(t\right)={\hat{X}}^{\left(1\right)}\left(t\right)+1_{\left\{P(+)\geq P(-)\right\}}{\hat{\varepsilon }}^{\left(1\right)}\left(t\right)-1_{\left\{P(+)<P(-)\right\}}{\hat{\varepsilon }}^{\left(1\right)}\left(t\right)\\ {\hat{Y}}^{\left(0\right)}\left(t\right)={\hat{Y}}^{\left(1\right)}{(t)-\hat{Y}}^{\left(1\right)}\left(t-1\right)\\ {\hat{X}}^{\left(1\right)}\left(t\right)=-15.5702e^{-0.0698t}+16.5702\\ {\hat{\varepsilon }}^{\left(1\right)}\left(t\right)=-0.1915e^{-0.1871t}+0.1915\\ P=\left[\begin{array}{cc} 4/5 & 1/5\\ 1/4 & 3/4 \end{array}\right] \end{array}\right.$$

The specific prediction results obtained are detailed in Table 5. The prediction curves of OPC in the acid rain environment for the two different models were plotted to compare their prediction accuracies visually (Figure 10). The results showed that the maximum relative error of the relative dynamic elastic modulus for the OPC group was 4.28% and that the average error was 1.79% in the prediction results based on the GM(1,1) model. The GM(1,1)-Markov model predicted a maximum relative error of 1.72% and an average error of 0.67%. This result revealed that the addition of Markov chains significantly improved the prediction accuracy of the model. As shown in Figure 10, the GM(1,1)-Markov model prediction curve changed smoothly because the Markov chain was able to fully consider the effect of residuals on the prediction results. OPC was predicted to have broken down to the point of failure by the GM(1,1)-Markov model before approaching 180 days. However, the GM(1,1) model’s prediction results revealed OPC was just close to failure. As the original data were tested every 20 days, the predictions of the GM(1,1)-Markov model were more consistent with the facts.

After confirming that the GM(1,1)-Markov model had a high prediction accuracy, the remaining life of each group of specimens was predicted by the model to further understand the damage deterioration trend of fiber concrete. The results are shown in Figure 11. The incorporation of multi-scale hybrid fibers improved the service life of concrete in the acid rain environment with the magnitude of service time. The service time of concrete at each mixture proportion from largest to smallest is S2B0.1 > S2B0.2 > S1B0.1 > S1B0.2 > OPC. The specific times are 322 d, 280 d, 230 d, 206 d, and 178 d.

## 5. Conclusions

In this paper, the durability damage deterioration law of fiber concrete under acid rain corrosive environment is investigated by both macroscopic and microscopic tests, and the service life is predicted by using a GM(1,1)-Markov model. Based on the tests and analysis, the following conclusions can be drawn.


(1)Incorporation of steel fibers and basalt fibers can play an inhibiting role in the deterioration of damage to concrete under the effect of acid rain corrosion. The best effect is achieved when the volume dose of SF is 2% and the volume dose of BF is 0.1%.(2)The microscopic test results show that the fibers share the expansion stress caused by corrosion products to a certain extent, improve the pore structure, reduce the thickness of ITZ, make the concrete micro-structure more dense, and enhance the resistance to acid rain corrosion.(3)The GM(1,1)-Markov model has a high prediction accuracy and can effectively predict the damage deterioration trend of fiber concrete, which provides a good theoretical basis for the repair and reinforcement of concrete in acid rain areas.


## Figures and Tables

**Figure 1 materials-14-06326-f001:**
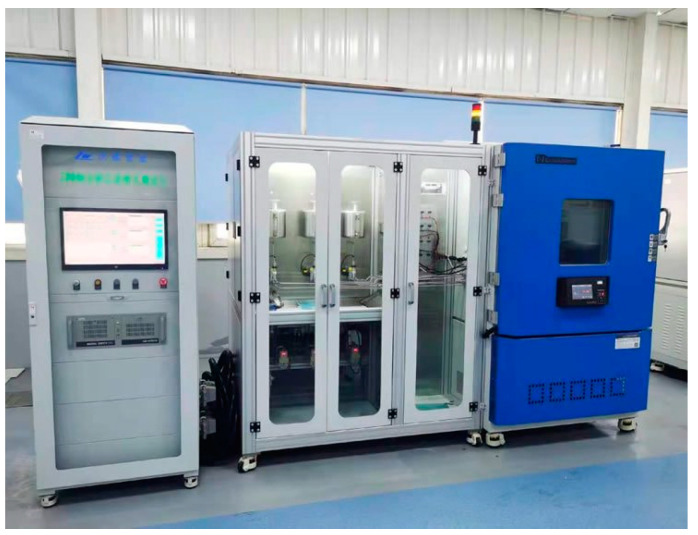
Indoor cycle test chamber.

**Figure 2 materials-14-06326-f002:**
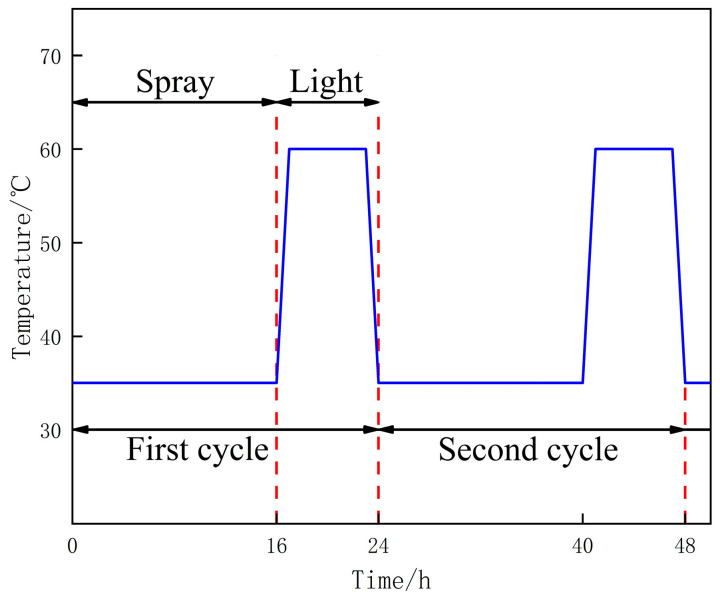
Spray-light corrosion cycle system diagram.

**Figure 3 materials-14-06326-f003:**
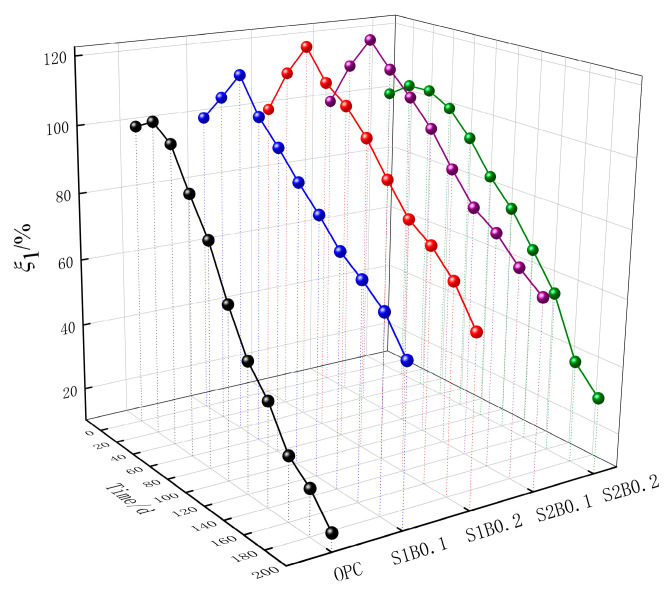
Time variation curve of compressive strength evaluation parameters.

**Figure 4 materials-14-06326-f004:**
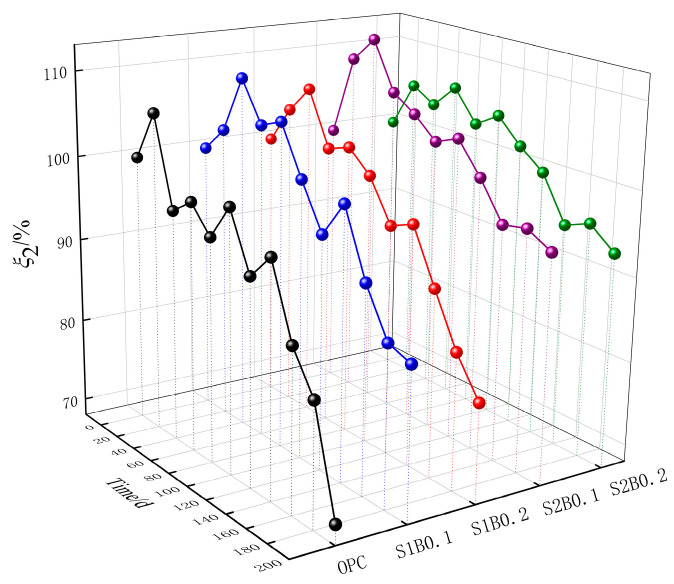
Time variation curve of relative mass evaluation parameters.

**Figure 5 materials-14-06326-f005:**
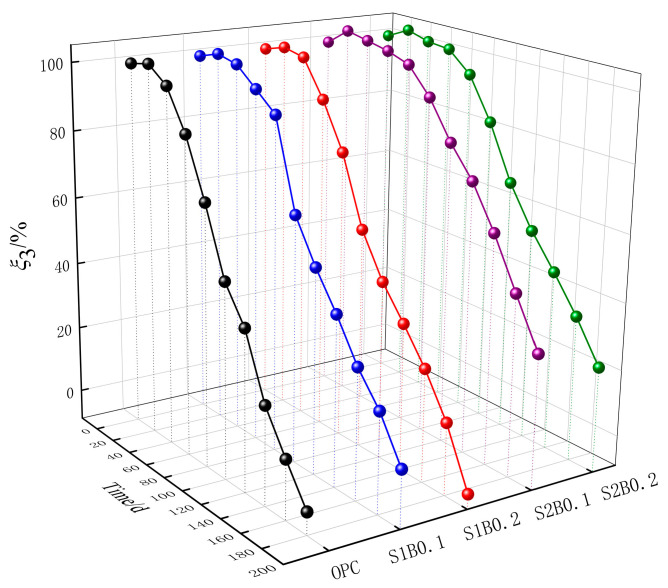
Time variation curve of relative dynamic elastic modulus evaluation parameters.

**Figure 6 materials-14-06326-f006:**
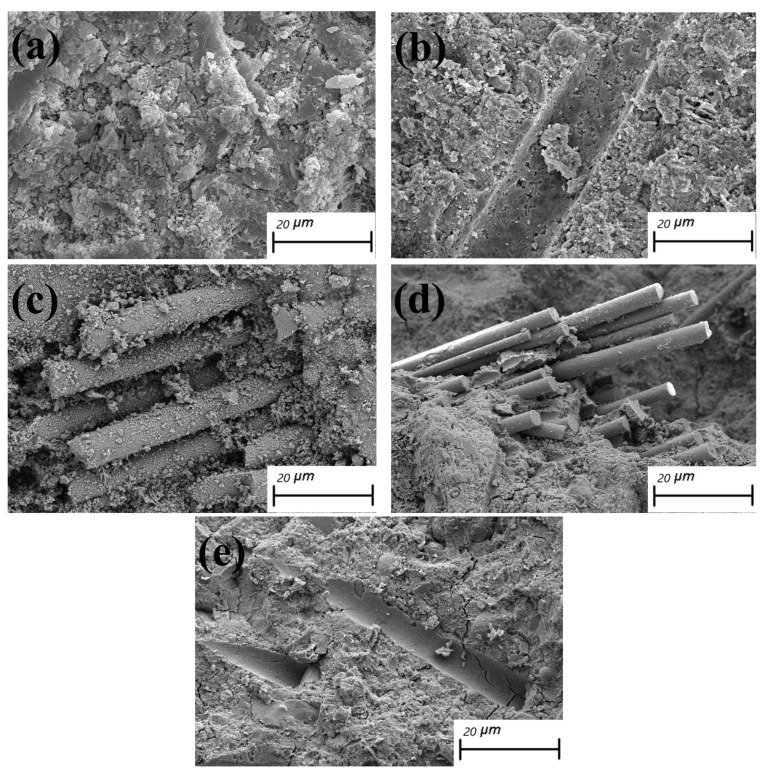
SEM images of five concrete: (**a**) OPC (**b**) S1B0.1 (**c**) S1B0.2 (**d**) S2B0.1 (**e**) S2B0.2.

**Figure 7 materials-14-06326-f007:**
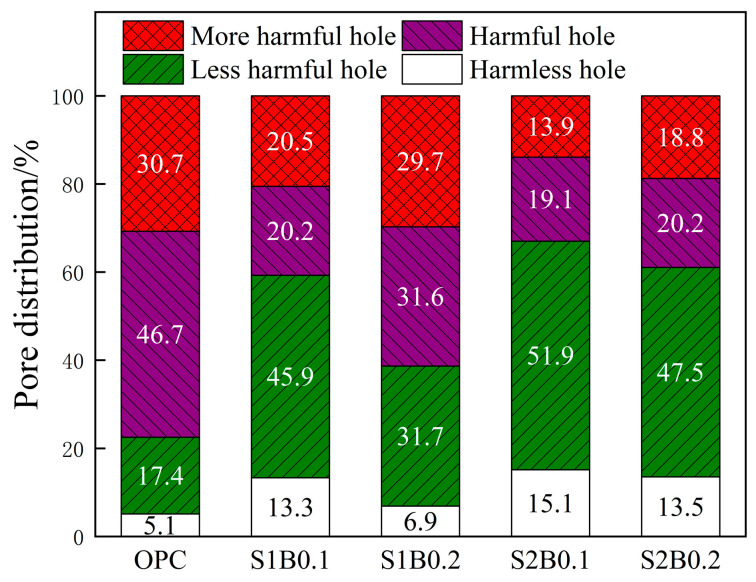
Pore distribution of fiber concrete under an acid rain environment.

**Figure 8 materials-14-06326-f008:**
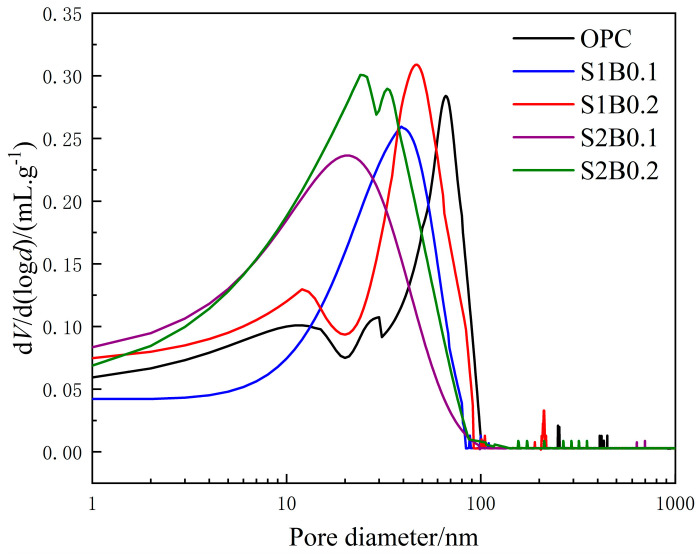
Pore size differential distribution curve of fiber concrete under an acid rain environment.

**Figure 9 materials-14-06326-f009:**
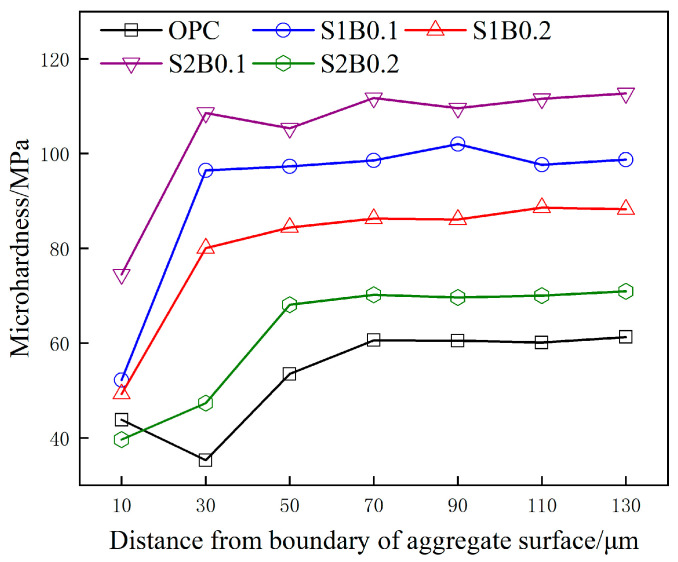
Microhardness curve of fiber concrete under an acid rain environment.

**Figure 10 materials-14-06326-f010:**
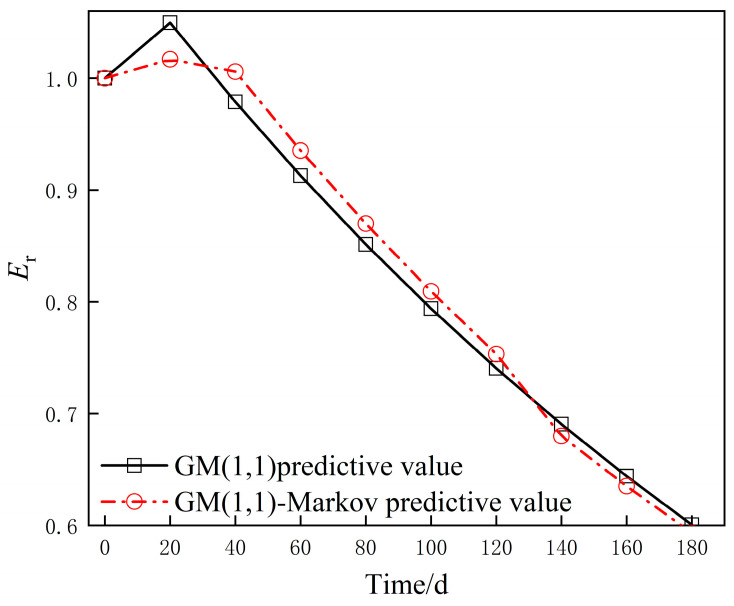
Model prediction comparison.

**Figure 11 materials-14-06326-f011:**
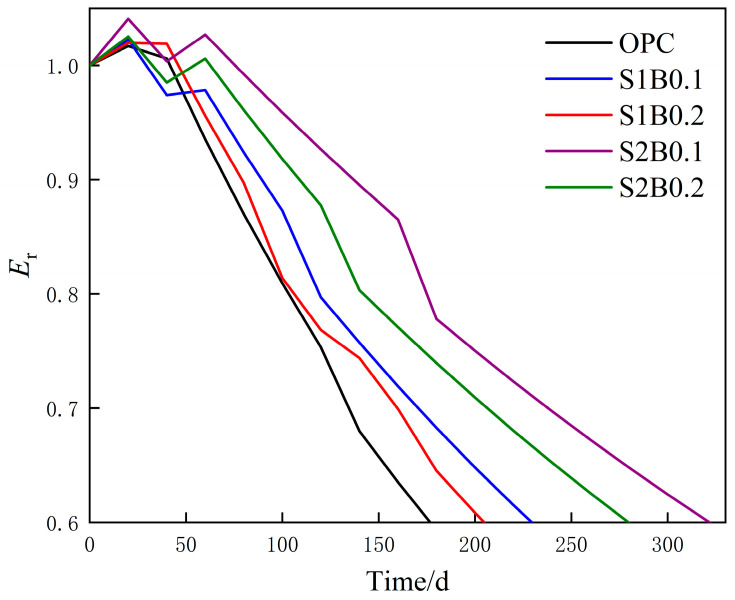
Remaining life curve of each group based on the GM(1,1)-Markov model.

**Table 1 materials-14-06326-t001:** Chemical composition of cementitious materials (w/%).

Material	Al_2_O_3_	CaO	K_2_O	IL	MgO	MnO	SO_3_	Fe_2_O_3_	SiO_2_
Cement	9.2	50.5	1.2	1.8	3.3	0.3	2.1	4.2	26.5
Fly ash	31.2	6.0	1.5	2.2	1.1	0.2	0.6	6.5	48.5

**Table 2 materials-14-06326-t002:** Main physical property parameters of fiber.

Type	Length/mm	Diameter/μm	Elastic Modulus/GPa	Tensile Strength/MPa	Density/kg·m^−3^	Shape
SF	30	500	200	1270	7800	Wavy
BF	12	20	100	4500	2700	Monofilament dispersion

**Table 3 materials-14-06326-t003:** Fiber concrete mix proportion.

Mix	OPC	S1B0.1	S1B0.2	S2B0.1	S2B0.2
Cement/(kg·m^−3^)	400	400	400	400	400
Sand/(kg·m^−3^)	633	633	633	633	633
Stone/(kg·m^−3^)	1167	1167	1167	1167	1167
Water/(kg·m^−3^)	200	200	200	200	200
Fly ash/(kg·m^−3^)	100	100	100	100	100
Superplasticizer/(kg·m^−3^)	0.65	0.65	0.65	0.65	0.65
SF/% (by volume fraction)	/	1.0	1.0	2.0	2.0
BF/% (by volume fraction)	/	0.1	0.2	0.1	0.2
Compressive strength/MPa	44.2	48.3	47.9	51.2	48.5

**Table 4 materials-14-06326-t004:** Chemical composition of the simulated acid rain solution (mol/L).

pH Value	H^+^	SO_4_^2−^	NH_3_^+^	NO_3_^−^
3.0	1.0 × 10^−3^	1.37 × 10^−3^	2.0 × 10^−4^	1.0 × 10^−3^

**Table 5 materials-14-06326-t005:** Prediction results of OPC group.

Time/d	Raw Data	GM(1,1)		${\hat{\varepsilon }}^{\left(0\right)}(t)$	P(x)	GM(1,1)-Markov	
		Fitting Value	Relative Error			Fitting Value	Relative Error
0	1.0000	1.0000	0.00%	0.0000	+	1.0000	0.00%
20	1.0065	1.0496	4.28%	0.0431	-	1.0169	1.03%
40	0.9889	0.9788	1.02%	0.0101	+	1.0059	1.72%
60	0.9426	0.9128	3.16%	0.0298	+	0.9353	0.78%
80	0.8752	0.8513	2.73%	0.0239	+	0.8699	0.60%
100	0.7966	0.7939	0.33%	0.0027	+	0.8094	1.61%
120	0.7559	0.7404	2.05%	0.0155	+	0.7532	0.36%
140	0.6815	0.6905	1.32%	0.0090	-	0.6799	0.24%
160	0.6350	0.6439	1.41%	0.0089	-	0.6351	0.02%
180	0.5910	0.6005	1.61%	0.0095	-	0.5932	0.38%

## Data Availability

The data that support the findings of this study are available from the corresponding author upon reasonable request.

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
