# Peer review of "Research on Damage and Deterioration of Fiber Concrete under Acid Rain Environment Based on GM(1,1)-Markov"

_materials, 2021, doi:10.3390/ma14216326_

Round 1

Reviewer 1 Report

This document illustrates a research programme carried out on the durability. This is one of the most important current issue under investigation not only for materials but also for civil engineering. The authors describe a challenge investigation, very interesting and hard-working in terms of durations and procedures to be followed. Conclusions are aligned to the obtained results.

Author Response

The author thanks the reviewer for the attention and kind suggestion! We appreciate so much for the warm and encouraging comments from you! In the revised manuscript, the quality of the images has been improved at the review-back stage, and the images have become clearer. In addition, some writing and expression errors have been corrected. In the section 2. Materials and Methods, the experimental methods and some test procedures were added to make the paper more logical as a whole. 

Reviewer 2 Report

Title: Research on damage and deterioration of fiber concrete under acid rain environment based on GM(1,1)-Markov

materials-1413808

The aim of this work is to study the durability performance of hybrid fiber reinforced concrete under acid rain environment. The abstract briefly introduces the subject and briefly describes the issues presented in the work. The introduction describes the research area well. The adopted methodology is correct. The conclusions are clear and are directly derived from the research carried out. There are a few things that should be improved as listed below.

I think it would be worth making some adjustments to the work:

  1. I would like to ask you to improve the quality of the drawings (increase DPI): 2, 3, 4, 5, 7, 8, 9, 10, 11.
  2. How were the numerical values in Equations 1, 2, and 3 determined?

The article after minor corrections, in my opinion, is suitable for publication in the Materials journal.

Author Response

Thanks a lot for your valuable question! Please see the attachment,the document is a total of 20 pages, of which pages 1-2 are the response notes and pages 3-20 are the revised article, and the text modifications are marked in red.

Reviewer 3 Report

The goal of this manuscript is attempting to quantitatively evaluate the durability damage deterioration law of fiber concrete under acid rain corrosive environment is investigated by both macroscopic and microscopic tests, and the service life is predicted by using GM(1,1)-Markov model. However, this manuscript lacks deep analysis and is not well-structured, and the evidence listed seems to be inadequate to support their opinion. Some reasons for rejecting the acceptance of the manuscript to the journal of Materials are given below.

  • It is not understandable why the damage parameters in Eqs. (1) to (3) calculated with different formulas. Why do we need to use 0.75, 0.25, 0.05, 0.95, 0.60, and 0.40? And why concrete should be considered when the damage parameters are less than 0?
  • What is the sample size for mercury piezometer? How to control the sample size after 200 days of environmental loading? How many samples were used? Can this represent the material nature?
  • For preparing the samples for the microhardness test, the reviewer couldn’t imagine how to precisely cut and polish the samples after 200 days of environmental loading without damage the samples considering the size of the samples.
  • Section 3.1: There is no discussion of why we have the following results: S2B0.2>S1B0.2>S1B0.1>S2B0.1.
  • Section 3.3: There is no discussion of why we have the following results: S2B0.1>S2B0.2>S1B0.1>S1B0.2.
  • Section 3.3: The claim by the authors that “The order of the magnitude of sensitivity of the three durability evaluation parameters of fiber concrete under acid rain corrosion was ξ3>ξ1>ξ2” is not a fair comparison because the calculations of the parameters are not the same. See Eqs. (1) to (3).
  • Section 3.4: The reviewer can’t follow the authors’ claims. There are too many speculations without evidence. We are just seeing about 80x60 um samples, in which comparing crack width is very difficult.
  • Section 3.6: The reviewer can’t follow the authors’ claims regarding 68 um and 27 um.
  • First conclusion: We can expect that the samples with higher compressive strength will result in higher durable characteristics.
  • Second conclusion: We can’t tell the synergistic effect in this study because there was no experimental case with single fiber addition.
  • Table 2: Tensile strength of steel fiber is generally greater than that of basalt fiber.
  • Section 2.2: It is not clear how many samples were test per group and each test date, and whether the authors averaged or not.
  • Table 3: The title of the table is not correct. ‘S21B0.2’ is a typo. The term ‘stone’ is not a correct expression. It is not clear whether the fiber contents are in volume or weight.
  • Minor issues: i) The term MIP in cementitious society is generally used for Mercury Intrusion Porosimetry or Mercury Porosimetry; ii) the unit of fiber diameter in Table 2 should be checked; iii) in page 12, aggregate-cement interface should be modified as aggregate-cement paste interface

Author Response

Thanks a lot for your valuable question! Please see the attachment,the document is a total of 23 pages, of which pages 1-5 are the response notes and pages 6-23 are the revised article, and the text modifications are marked in red.

Reviewer 4 Report

In this work, authors deal with the study of fiber concrete performances after the incorporation of steel and basalt fiber volumes. In particular, their investigated the durability under acidic rain corrosion by macroscopic and microscopic perspectives and they predicted the service life by using the GM(1,1)-Markov model.

Overall, the manuscript is very interesting and It has a good scientific soundness. However, at this state, it presents many lacks that must be addressed by considering the following major revisions:

1.How were the chemical components of the cementitious material in Table 1 collected?

2. the same for the main properties of the steel fibers (SF) and basalt fibers (BF) are in Table 2. How were the mechanical properties measured?

3.table 3 should be placed under table 2. Now there is just a “3” without meaning.

4.the quality of figure 2 should be improved. Also, it is strongly recommended to describe the meaning of figure 2 and the different steps of acidic rains simulation in the experimental section.

5.the numbers of equations 1,2 and 3 should be the same and in the same position.

6.What are the experimental conditions for SEM analysis (e.g. beam energy, working distance, etc.)? Were the samples metal sputtered before experiments to avoid electron charging and to have better quality images?

7.when discussing data in figure 3, authors report that “However, as the age of erosion increased, the large amount of generated expansions produced microcracks by extrusion within the concrete, thus causing a decrease in compressive strength”. This point is not clear, please discuss better.

8.please revise paragraph 3.2 “which may be due to the fact that the physical crystallization destruction ability of spray-light has not yet caused the fiber concrete to exhibit quality loss behaviors, including dissolution precipitation”. It is not very clear why some sentences are bold style?

9.Axes in figure 3, 4 and 5 can not be clearly read. Please improve the quality. For example is almost impossible to read time scale.

10. Figure 7 and actually all the different graphs in the manuscript. Please revise and improve quality for better clarity. In figure 7, for example, the names of samples are cut.

Author Response

Thanks a lot for your valuable question! Please see the attachment,the document is a total of 22 pages, of which pages 1-4 are the response notes and pages 5-22 are the revised article, and the text modifications are marked in red.

Round 2

Reviewer 3 Report

The reviewer would express gratitude for preparing detailed responses from the authors. However, the concerns raised by the reviewer were still unresolved. Unfortunately, the reviewer couldn’t agree with the acceptance of this article. Some reasons for rejecting the acceptance of the manuscript to the journal of Materials are given below.

Comment 4: The reviewer asked to add critical discussions, but the authors still reported the simple findings.

Comments 1 and 5: Simply following the parameters suggested in Chinese standards is not persuasive.

Comment 6: Observing very limited areas can’t represent overall material structures.

Comment 8: As agreed by the authors, samples with higher compressive strength will result in higher durable characteristics. Therefore, the first conclusion might not be correct.

Reviewer 4 Report

The manuscript can be accepted now.

This manuscript is a resubmission of an earlier submission. The following is a list of the peer review reports and author responses from that submission.